# Application of a Control Scheme Based on Predictive and Linear Strategy for Improved Transient State and Steady-State Performance in a Single-Phase Quasi-Z-Source Inverter

**DOI:** 10.3390/s22072458

**Published:** 2022-03-23

**Authors:** Manuel Diaz-Bustos, Carlos R. Baier, Miguel A. Torres, Pedro E. Melin, Pablo Acuna

**Affiliations:** 1Department of Electrical Engineering, Universidad de Talca, Camino a los Niches Km. 1, Curicó 3344158, Chile; mandiaz@utalca.cl (M.D.-B.); pablo.acuna@utalca.cl (P.A.); 2Instituto de Ciencias de la Ingeniería, Universidad de O’Higgins, Rancagua 2841959, Chile; miguel.torres@uoh.cl; 3Department of Electrical and Electronic Engineering, Universidad del Bío-Bío, Concepción 4051381, Chile; pemelin@ubiobio.cl

**Keywords:** FCS-MPC, single-phase quasi impedance source inverters, qZSI, hybrid control schemes, single-phase inverters

## Abstract

Z and quasi-Z-source inverters (Z/qZSI) have a nonlinear impedance network on their dc side, which allows the system to behave as a buck–boost converter in their outputs. The challenges derived from the qZSI topology include (a) the control of the voltage and current on its nonlinear impedance network, (b) the dynamic coupling between the ac and dc variables, and (c) the fact that a unique set of switches are used to manage the power at dc and ac side of the system. In this work, a control scheme that combines a PWM linear control strategy and a strategy based on finite control state model predictive control (FCS-MPC) is proposed. The linear approach works during steady state, while the FCS-MPC works during transient states, either in the start-up of the converter or during sudden reference changes. This work aims to show that the performance of this control proposal retains the best characteristics of both schemes, which allows it to achieve high-quality waveforms and error-free steady state, as well as a quick dynamic response during transients. The feasibility of the proposal is validated through experimental results.

## 1. Introduction

Power inverters are currently used in a wide range of industrial applications, and will be essential for the operation of future distribution networks [1,2]. Depending on users’ needs, different types of inverters can be used to feed different types of load, such as voltage source inverters (VSI), current source inverters (CSI), and impedance source inverters (ZSI, qZSI, and similar) [3,4,5].

Quasi-Z source inverters (QZSIs) combine, in one stage, a buck–boost converter on the dc side and a dc–ac inverter [6,7]. These quasi-Z source topologies are a different alternative to the classical topologies (voltage and current source) known fifteen years ago [8,9]. In fact, their buck–boost feature differentiates them from VSIs, which behave as buck inverters in their ac outputs, and also from the CSIs, which behave as boost inverters on the load side [10,11].

To behave as a buck–boost inverter, ZSI and qZSIs need a nonlinear network on their dc side to be able to switch between two states: non-shoot-through state (*n*STS) and shoot-through state (STS). This nonlinear network and its operating states inherently behave as a non-minimum phase system [8,12]. Control strategies for these types of converters can be a significant challenge, given the following factors: (a) control of the direct current and alternate current variables has to be performed with the same set of power switches, (b) the dynamics of the dc side and the ac side are completely different, and (c) the dc side behaves as a non-minimum phase system [8,13].

Some control strategies to achieve correct operation during steady state have already been developed. Among these strategies are those that use linear controllers and sinusoidal pulse width modulation (SPWM). These PWM linear strategies have revolved around directly or indirectly controlling the dc voltage at the input of the inverter bridge, minimizing at the same time the currents into the qZSN dc side inductances [8,14,15].

In relation to the different dynamics between dc and ac variables of ZSIs, it has been shown that nonlinear control strategies might allow faster responses under sudden reference and load changes in these converters. Among these strategies are fuzzy control [16,17], sliding-mode control [18,19], neural network control [20,21], and finite control set model predictive control (FCS-MPC) [22,23,24,25]. The latter strategy is of more interest for this work, because of its simplicity of implementation and its fast dynamic behavior. On the other hand, linear PWM control achieves low harmonic distortion and zero error in a steady state; however, obtaining fast and similar dynamic responses at any operating point is not a straightforward task. In contrast, an FCS-MPC strategy easily allows to achieve an optimal and rapid response under virtually any reference change; however, the resultant harmonic distortion in a steady state is higher than the one obtained with linear PWM schemes [26].

The use of hybrid strategies that combine the advantages of different control schemes has been carried out before in power electronic applications [27,28,29]. In this regard, this work aims to design a hybrid control scheme for a single-phase quasi-Z-source inverter (SP-qZSI) that takes advantage of desired features of both predictive and linear control. In this novel approach, a linear PWM and an FCS-MPC strategy operate together using an algorithm that synchronizes their operation. Therefore, the proposed controller alternates between each strategy when there is a reference or disturbance change. The resulting strategy inherits desired characteristics of both control schemes when used independently, which are good performance in steady state (low voltage and current distortion) and fast response during transients.

This article is organized as follows. In Section 2, the fundamental aspects of quasi-Z-source inverters are presented. Then, in Section 3, the linear PWM and FCS-MPC strategies are presented and introduced as the control modes that will be implemented in this proposal. In Section 4, the algorithm that enables the synchronized operation of both strategies is described. In Section 5, the experimental results that validate the controller operation are presented. Finally, in Section 6, the main conclusions of this work are summarized.

## 2. The Single-Phase Quasi-Z-Source Inverter

A single-phase quasi-Z-source inverter (SP-qZSI) fed by a dc voltage source is composed of a nonlinear network—called quasi-Z source network (qZSN)—on the dc side and an H-bridge inverter. This structure, which could feed an RL load (among other load types) on the ac side, is shown in Figure 1a. The inverter takes power from the dc voltage source ($V_{in}$) and, depending on its operating state, the energy circulates through the qZSN or it is stored by the reactive elements. The energy available in dc source and in the qZSN is then sent to the ac load through the H-bridge inverter using a modulation and/or a control technique.

Regarding the operation of the dc side of the qZSI, both modes of operation can be considered: *n*STS and STS. The circuits equivalent to these two operating modes are shown in Figure 1b,c, respectively. In *n*STS, the inverter can take on three tasks: (i) positively energize the load, (ii) negatively energize the load, or (iii) apply a null state (in which capacitors and inductors are loaded), keeping the diode $D_{1}$ in conduction. When the inverter in Figure 1 is in *n*STS, from the ac side, it operates similarly to a conventional VSI. On the other hand, in STS, both upper and lower switches of the inverter legs close simultaneously and cause the diode $D_{1}$ to stop conducting, at the same time allowing a small discharge in the qZSN as long as it remains in STS. The control schemes proposed for the dc side should consider both states (*n*STS and STS) to control the variables of the system. For a better understanding of its function, the inverter model from the dc side is described below.

**Figure 1 sensors-22-02458-f001:**
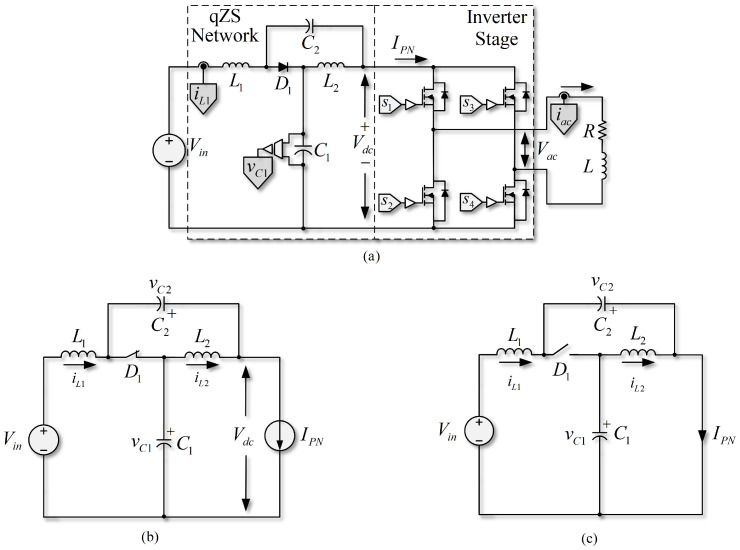
Quasi-Z-source inverter. (**a**) qZSI with RL load; (**b**) dc equivalent non-shoot-through state circuit; (**c**) dc equivalent shoot-through state circuit.

### Model of the dc Side at the qZSI

Figure 1b,c show that if the voltage drop in diode $D_{1}$ of the *n*STS circuit (Figure 1b) is disregarded, it is possible to describe the dynamics of the currents in the inductances of the qZSN as
(1)$$\frac{di_{L1}}{dt}=\frac{1}{L_{1}}\left(V_{in}-v_{C1}\right),$$
(2)$$\frac{di_{L2}}{dt}=-\frac{v_{C2}}{L_{2}},$$
where $V_{in}$ is the input dc voltage, $v_{C1}$, $v_{C2}$ are the voltages in the capacitors $C_{1}$ and $C_{2}$, and $i_{L1}$, $i_{L2}$ are the currents in the inductors $L_{1}$ and $L_{2}$. Meanwhile, the capacitors voltages dynamics of the qZSN are given by
(3)$$\frac{dv_{C1}}{dt}=\frac{1}{C_{1}}\left(-I_{PN}+i_{L1}\right),$$
(4)$$\frac{dv_{C2}}{dt}=\frac{1}{C_{2}}\left(-I_{PN}+i_{L2}\right).$$

By analyzing the circuit of Figure 1c that represents the STS, the equations that describe the behavior of the qZSN variables can be written as follows:(5)$$\frac{di_{L1}}{dt}=\frac{1}{L_{1}}\left(V_{in}-v_{C2}\right),$$
(6)$$\frac{di_{L2}}{dt}=\frac{v_{C1}}{L_{2}},$$
(7)$$\frac{dv_{C1}}{dt}=-\frac{i_{L2}}{C_{1}},$$
(8)$$\frac{dv_{C2}}{dt}=-\frac{i_{L1}}{C_{2}}.$$

The binary variable $S_{ST}$ (Equation 9) is introduced to define the state (*n*STS and STS) in which the inverter is operating.
(9)$$S_{ST}\;=\;\left\{\begin{array}{c} 1\;if\;\;\;\mathrm{STS}\;\\ 0\;if\;n\mathrm{STS}, \end{array}\right.$$
with the use of (Equation 9) it is possible to write a conditional model of the dc side that depends on variable $S_{ST}$, which is written as
(10)$$\left\{\begin{array}{c} \frac{di_{L1}}{dt}\;=\;\frac{\left(1-S_{ST}\right)\left(V_{in}-v_{C1}\right)+S_{ST}\left(V_{in}-v_{C2}\right)}{L_{1}}\\ \frac{di_{L2}}{dt}\;=\;\frac{\left(1-S_{ST}\right)\left(-v_{C2}\right)+v_{C1}S_{ST}}{L_{2}}\\ \frac{dv_{C1}}{dt}\;=\;\frac{\left(1-S_{ST}\right)\left(i_{L1}-I_{PN}\right)-i_{L2}S_{ST}}{C_{1}}\\ \frac{dv_{C2}}{dt}\;=\;\frac{\left(1-S_{ST}\right)\left(i_{L2}-I_{PN}\right)-i_{L1}S_{ST}}{C_{2}} \end{array}\right.$$
where $L_{1}$, $L_{2}$, $C_{1}$, and $C_{2}$ are the inductances and capacitances of their respective elements in the qZSN, and variables $i_{L1}$, $i_{L2}$, $v_{C1}$, and $v_{C2}$ are the inductors currents and the capacitors voltages at the dc side nonlinear network.

From the model described in (Equation 10), it is possible to obtain the average behavior when including both states of operation based on the duty cycle *d* of STS. The resulting model is
(11)$$\left\{\begin{array}{c} \frac{di_{L1}}{dt}\;=\;\frac{\left(1-d\right)\left(V_{in}-v_{C1}\right)+d\left(V_{in}-v_{C2}\right)}{L_{1}}\\ \frac{di_{L2}}{dt}\;=\;\frac{\left(1-d\right)\left(-v_{C2}\right)+v_{C1}d}{L_{2}}\\ \frac{dv_{C1}}{dt}\;=\;\frac{\left(1-d\right)\left(i_{L1}-I_{PN}\right)-i_{L2}d}{C_{1}}\\ \frac{dv_{C2}}{dt}\;=\;\frac{\left(1-d\right)\left(i_{L2}-I_{PN}\right)-i_{L1}d}{C_{2}} \end{array}\right.$$

Taking into account a steady-state operation through Equation (Equation 11), it is possible to find the gain $G_{v_{c1}}$ of the capacitor voltage $C_{1}$ related to the input voltage $V_{in}$ in terms of *d* for STS, i.e.,
(12)$$G_{v_{c1}}\;=\;\frac{v_{c1}}{V_{in}}\;=\;\left(\frac{1-d}{1-2d}\right).$$

In the same way, it is possible to find a voltage gain $G_{v_{c2}}$ related to the second capacitor as
(13)$$G_{v_{c2}}\;=\;\frac{v_{c2}}{V_{in}}\;=\;\left(\frac{d}{1-2d}\right).$$

Finally, the dc input current gain $G_{i}$ can be written as
(14)$$G_{i}\;=\;\frac{i_{L1}}{I_{PN}}\;=\;\frac{i_{L2}}{I_{PN}}\;=\;\left(\frac{1-d}{1-2d}\right).$$

These gains indicate that it is possible to fine-tune the voltages and currents on the dc side of the inverter in steady state by using a linear control strategy. However, the modulation needed will limit the achievable values of these gains, as well as the values of the duty cycles, as shown in [30]. It is important to note that the proposed mathematical model and expressions are derived from previous research and have already been used for applications where modeling is essential to control the system [18,31]. In this context, this work proposes to add a predictive strategy for transient conditions.

## 3. Linear and Predictive Control Schemes for a SP-qZSI

This section briefly describes the two control schemes that will be used as control modes in the proposed hybrid control strategy.

### 3.1. PWM Linear Control in SP-qZSI

Figure 2 shows the PWM linear control scheme in the SP-qZSI. Given that it is essential to control the dc voltage in the capacitors, limiting at the same time the currents in the qZSN inductances of the SP-qZSI, a valid control scheme that the converter can use for the dc side is a double-loop PI control. The external loop of the control is used to regulate the capacitor voltage ($v_{C1}$). In contrast, the internal feedback is used to limit the inductor current ($i_{L1}$), as shown in Figure 2b, in a similar fashion as in [32]. Double-loop controllers (or cascade) can achieve good regulation of the external loop reference (qZSN capacitors voltage) and both limit and stabilize the values of the internal loop variables (qZSN inductors current). One interesting advantage of the double-loop configuration is that it allows the control strategy to be more robust [32].

The tuning of both controllers, first the current loop controller, and then the voltage loop in the capacitor, can be performed by the procedure given in [33]. Then, control with stable responses is achieved within the operation range of the converter, taking into account the parameters of the qZSN.

On the other hand, if output power control is required, it will be necessary to control the ac side current $i_{ac}$ of the inverter as well. However, in the case of ac variables, it is not possible to achieve error-free performance in steady state using PI controllers, since the pole in the origin of PI only allows error-free performance for dc variables [34,35]. A feasible alternative is to use dq or dq0 transforms, but that increases the number of PI controllers in single-phase systems, and it does not have good harmonic rejection capability [36,37]. On the contrary, when using a proportional-resonant (PR) controller, it is possible to control the ac load current $i_{ac}$ of the SP-qZSI, with good tracking and without error in steady state. The reason is that the PR satisfies the internal model principle (IMP) by including a dynamic model of sinusoidal signals in the expression of the controller [34]. The transfer function of a PR controller can be written as follows:(15)$$H_{PR}\left(s\right)\;=\;PR_{ac}\;=\;K_{Pr}\;+\;K_{R}\frac{s}{s^{2}+\omega_{0}^{2}}$$
where $K_{Pr}$ is the controller proportional gain, $K_{R}$ is the resonant gain, and $\omega_{0}$ is the fundamental frequency of the reference or system in which it should operate. If the set-point reference to the system is a sinusoidal with period $\frac{1}{\omega_{0}}$, this controller theoretically achieves an infinite gain at frequency $\omega_{0}$, which allows for an error-free follow-up in steady state by closing the control loop for the system on the ac side [10,34].

Both the PI and PR controller offer the advantage of achieving error-free follow-up of its references in steady state, so the scheme proposed in Figure 2 is adequate to control the converter in this state. However, the different dynamics of the ac and dc sides may produce undesirable effects on the response of this control scheme.

A control scheme that achieves better results from the point of view of the dynamics in the SP-QZSI is described below.

### 3.2. Predictive Control in the SP-qZSI

It is possible to use a finite control set model predictive control (FCS-MPC) strategy to improve the proposed converter’s dynamic behavior, since this kind of control method can consider all the nonlinearities in the system. The proposed control scheme considers a model that allows the prediction of the different future states of the converter by using information acquired in a present condition and all possible combinations of the switches in the topology. Valid switching states of SP-qZSI can be seen in Table 1.

The state to be applied is selected based on optimization of a cost function, in which a comparison between the references of one or several variables and the prediction is made [38].

A typical FCS-MPC scheme used in SP-qZSI is presented in Figure 3. The proposed strategy controls both the inductor current and the capacitor voltages of the qZSN simultaneously. The ac current is also managed in the same strategy. The proposed algorithm of the predictive scheme is shown in Figure 4. It is important to note that regulation of voltages $v_{C1}$ and $v_{C2}$ can be achieved by regulating only $v_{C1}$, and regulation of currents $i_{L1}$ and $i_{L2}$ can be achieved by regulating only $i_{L1}$; then, only the future states of $v_{C1}$ and $i_{L1}$ need to be predicted by the proposed strategy.

Considering the following equation that express the relationship between voltages,
(16)$$v_{C2}\;=\;v_{C1}-V_{in};$$
then, from the dynamic model described in the previous section, the proposed strategy can use the following prediction equations:(17)$$i_{L1}\left(k+1\right)\;=\;\frac{T_{s}}{L_{1}}\left(\left(1-S_{ST}\right)\left(V_{in}-v_{C1}\left(k\right)\right)\;+\;S_{ST}v_{C1}\left(k\right)\right)\;+\;i_{L1}\left(k\right)$$
and
(18)$$v_{C1}\left(k+1\right)\;=\;\frac{T_{s}}{C_{1}}\left(\left(1-S_{ST}\right)\left(i_{L1}\left(k\right)-I_{PN}\left(k\right)\right)-S_{ST}i_{L1}\left(k\right)\right)\;+\;v_{C1}\left(k\right),$$
where $T_{s}$ is the sampling period of the implementation, $v_{C1(k)}$ and $i_{L1(k)}$ are the voltage in the capacitor $C_{1}$ and the inductor current $L_{1}$ in the discrete-time k. The variable $S_{ST}$ is the same as defined in Equation (Equation 9). Since the parameter values of inductances $L_{1}$ and $L_{2}$ in the qZSN are the same, the prediction of the current $i_{L1}$ is valid for both inductances. In addition, the prediction of the voltage $v_{C1}$ is enough to manage the voltage in both capacitors of the qZSN.

If a second prediction step is considered to implement an extended-horizon MPC, the scheme will also take the following pair of equations into account:(19)$$i_{L1}\left(k+2\right)\;=\;\frac{T_{s}}{L_{1}}\left(\left(1-S_{ST}\right)\left(V_{in}-v_{C1}\left(k+1\right)\right)\;+\;S_{ST}v_{C1}\left(k+1\right)\right)\;+\;i_{L1}\left(k+1\right)$$
and
(20)$$v_{C1}\left(k+2\right)\;=\;\frac{T_{s}}{C_{1}}\left(\left(1-S_{ST}\right)\left(i_{L1}\left(k+1\right)-I_{PN}\left(k+1\right)\right)-S_{ST}i_{L1}\left(k+1\right)\right)\;+\;v_{C1}\left(k+1\right),$$

The cost function necessary for controlling the dc side variables must consider the voltages and currents in the QZSN. The ac reference and the STS and nSTS can be considered for generating the dc inductance current reference. Thus, the required reference can be written as
(21)$$i_{Lref}\left(k\right)\;=\;\left(1-S_{ST}\right)h_{SW}\left(k\right)i_{acref}\left(k\right),$$
where $h_{SW}\left(k\right)$ can take values {1,0,−1} and corresponds to the switching taken by the single-phase H-bridge in nSTS, where $i_{acref}\left(k\right)$ is the current reference of the inverter ac side (load reference).

Since the system has a dc side and an ac side, for obtaining the optimal switching state the proposed strategy overall cost function is defined as the sum of a cost function for the qZSN (dc side) and a cost function of the load current (ac side), in the following way:(22)$$g\;=\;g_{dc}\;+\;g_{ac}.$$

The cost function on the dc side, can be expressed as
(23)$$g_{dc}\;=\;\sum_{l=1}^{N_{p}}\left(\lambda_{v}{\left(v_{C_{Ref}}-v_{c1}\left(k+l\right)\right)}^{2}\;+\;\lambda_{i}{\left(i_{L_{Ref}}-i_{L1}\left(k+l\right)\right)}^{2}\right)$$
where $\lambda_{v}$ and $\lambda_{i}$ are the weighting factors that will allow for faster or slower voltage control in the capacitor concerning the inductor’s current, and $N_{p}$ is the prediction horizon. A correct choice of the weighting factors will produce slower changes in the voltages of the capacitances in favor of obtaining more limited current in the qZSN inductances.

In order to implement the MPC of the ac side, the one-step prediction model for the load current $\left(i_{ac}\left(k+1\right)\right)$ is presented as
(24)$$i_{ac}\left(k+1\right)\;=\;\frac{T_{s}}{L}\left(\left(2v_{C1}-V_{in}\right)S_{f}-Ri_{ac}\left(k\right)\right)\;+\;i_{ac}\left(k\right),$$
and the cost function ($g_{ac}$) can be written as
(25)$$g_{ac}\;=\;\sum_{l=1}^{N_{p}}\lambda_{ac}{\left(i_{ac_{ref}}-i_{ac}\left(k+l\right)\right)}^{2}$$

With the dc and ac side variables prediction and the minimization of both cost functions, it is possible to find the switching state that allows an optimal tracking of the references or desired values. As the FCS-MPC scheme considers the nonlinearities in the system model, it will be able to provide optimal dynamic responses, mainly in the converter start-up or in case of significant changes in the references, which is not easy to guarantee by using a linear control scheme. Conversely, the operation of FCS-MPC in steady state has poorer performance indices than a PWM linear control scheme, mainly when operating with low sampling rates in the control system.

Given the difficulties in achieving fast dynamic responses under sudden reference changes and high-performance indices in the SP-qZSI simultaneously by using only one control scheme, it is proposed to use both described control schemes as control modes to obtain the best performance. As indicated before, an algorithm is required to coordinate the operation of both control modes (CM) depending on whether the system is in steady state or in transient state.

## 4. Alternating Control Modes in SP-qZSI

The proposed method in this article considers alternating between each control mode (CM) according to the working state of the converter. Since the regimes to be considered are steady and transient states, the CMs can be defined as steady control mode (SCM) and dynamic control mode (DCM), respectively.

To alternate between each CM, it is possible to define an alternating control mode algorithm (ACMA) which entails using a selection variable $\eta_{CM}$, as described in [27]. The following steps are required to design the ACMA: (a) selecting the control modes that will work on steady state and transient state independently, and (b) defining the criteria to switch between CMs.

The control modes have already been introduced, which are the PWM linear control strategy to operate as SCM, and the FCS-MPC to operate as DCM. Now, the criteria to switch between CMs is discussed below.

### 4.1. Switching Criteria of the Control Modes

#### 4.1.1. Basic Switching Criterion

The most basic definition of a CM selection (SCM or DCM) can be based on the tracking error of the least changing variable of the system. In the SP-qZSI, the capacitor voltage $v_{C1}$ of the qZSN is undoubtedly the slowest and most regular of the system. Therefore, in this converter, it is possible to use the difference between the voltage measured in the capacitor $v_{C1}$ and its reference $v_{C_{Ref}}$, i.e., $e_{v_{c}}\;=\;\left|v_{C_{Ref}}-v_{C1}\right|$, to define a selection flag $\eta_{CM}$, that is,
(26)$$\eta_{CM}\;=\;\left\{\begin{array}{c} 1,\;if\;\left|e_{v_{c}}\right|\leq \rho_{E}\\ 0,\;if\;\left|e_{v_{c}}\right|>\rho_{E} \end{array}\right.,$$
where $\rho_{E}$ is the set error limit. This means that if the error in the capacitor voltage control is lower than $\rho_{E}$, variable $\eta_{CM}$ will have a value of 1, so the system should operate in SCM, using the PWM linear control scheme. On the contrary, if the error is higher than $\rho_{E}$, the system should operate under DCM, which means that the system should work with FCS-MPC.

However, the selection criteria presented in (Equation 26) will face “bouncing” issues when it goes from DCM to SCM, or vice versa, because the variable may oscillate through the limit $\rho_{E}$ set up for a while, which would not be desirable.

#### 4.1.2. Improved Switching Criterion

To solve the bouncing issue that a basic criterion may encounter, a second criterion is proposed. In this second criterion, a hysteresis band for the error is introduced by adding a $\rho_{H}$ variable, which will serve as a countermeasure for the bouncing issue. When the error value is lower than $\rho_{E}$, and when it manages to make the switch from DCM to SCM, it remains in SCM until the error value does not exceed $\rho_{H}$, in which case the value would be higher than $\rho_{E}$.

This criterion may be written as
(27)$$\eta_{CM}\;=\;\left\{\begin{array}{c} 1,\;\;if\;\;\left\{\begin{array}{c} \left|e_{v_{c}}\right|\leq \rho_{E}\\ \left|e_{v_{c}}\right|\leq \rho_{H}\;\;\&\;\;\eta_{CM}=1 \end{array}\right.\\ 0,\;\;\;if\;\;\;\left|e_{v_{c}}\right|>\rho_{E} \end{array}\right.$$

Using this last criterion, a more adequate alternating control mode algorithm (ACMA) for SP-qZSI is proposed. The flow diagram of the proposed ACMA, and how it interacts with SCM, is shown in Figure 5. The ACMA is divided into two parts: (a) CM criteria and (b) CM selection. For the CM criteria, the ACMA verifies the voltage error value that comes from the SCM, and if this is lower than $\rho_{E}$, the selection flag will be equal to one. Conversely, if the error value is lower or equal to $\rho_{H}$, and at the same time the current flag $\eta_{CM}$ is equal to one, the value of this flag will continue to be one. Then, if the above criteria are not met, $\eta_{CM}$ will be equal to zero. Finally, the ACMA decides which strategy should operate in the CM selection section. For this, a commutation selection block will pass the operation from SP-qZSI to SCM (PWM linear control) or DCM (FCS-MPC) accordingly. It should be highlighted that the selection flag $\eta_{CM}$ itself can be used to activate or deactivate the PI and PR controllers of the SCM (as shown in Figure 5), which is crucial, because the PI error value should be zero during DCM, so there will be no increase in the controller integration. Additionally, when back to SCM, the PI and PR controllers will control smoothly if the error value is small.

### 4.2. Implementation Using ACMA in an SP-qZSI

The algorithm presented above allows for switching between two control strategies according to the operation regime under which the system is working. In Figure 6, the general control scheme with ACMA for SP-qZSI is presented. The implementation consists of four main stages. In the first stage, or power stage, sensing of the signals is carried out. In both control modes, it is necessary to measure $v_{C1}$, $i_{L1}$, and $i_{ac}$; thus, there are no differences in terms of implementation.

In the lower part of Figure 6, looking at the stage of the CMs, FCS-MPC and linear controllers are found. They both receive the measurements from the sensors installed in the power stage and generate binary switching signals from both CMs. The error $e_{v_{c}}$ is also calculated, which then is used in the ACMA, to provide the selection flag $\eta_{CM}$.

Using the selection flag $\eta_{CM}$, the corresponding multiplexors of the switching selection stage are activated. Then, proper command signals from either the SCM (PWM linear control) or the DCM (FCS-MPC) are selected. Both values of the ST signals and the gating signals of each switch reach the inverter through a logic addition.

## 5. Experimental Verification

To validate the proposal presented in this article, a low-power prototype was implemented (depicted in Figure 6). In this prototype, each control strategy was independently implemented. From the results, it was observed how the inverter operates with each strategy under both steady and dynamic conditions. The prototype was subjected to sudden changes in the DC voltage reference. Finally, these results were compared with those obtained for the inverter operating with the proposed scheme, in which both CMs operate simultaneously, activated by the ACMA.

The qZSN parameters in the prototype used in this work have been calculated according to the known indications [39]. The parameters and references applied to the practical experiences are described in Table 2. Table 3 shows the control scheme parameters used (MPC weighting factors and linear control gains), as well as the key frequencies considered for both schemes. It is essential to mention that the weighting factors of the FCS-MPC control have been found using guidelines from [40]. In addition, the gains of linear controllers (PI and PR) were tuned using methods given in [41,42].

Table 4 shows the parameters for the ACMA, which are crucial to avoid excessive transitions between control modes (bouncing). As a guideline to select them, $\rho_{E}$ is chosen as half of the maximum voltage oscillation value in the capacitor under steady state, and $\rho_{H}$ as its maximum oscillation value in steady state.

### 5.1. qZSI Operating under a PWM Linear Control

First, only the PWM linear control explained in Section 3—A will be considered, which is equivalent to forcing the system to operate under SCM at all times. Figure 7 shows that, in the face of a step change in the voltage reference of the capacitors from 40V to 65V, the controller is able to control the voltage adequately, that is, each capacitor voltage, as well as the load current. The capacitor voltage response includes a small overshoot and then the voltage is stabilized to its desired value, where the settling time is equivalent to two operating cycles. It is possible to see that the load current is barely affected when the change is performed, and that its lowest THDi is 5.1%. Additionally, a significant change in the inverter output voltage is observed.

On the other hand, Figure 8 shows that a step change in the opposite direction, that is from 65V to 40V, obtaining a slower dynamic response. This occurs since the capacitors should discharge their energy into a load which draws constant power and that mainly receives its energy from the qZSN inductances. In the waveform, it can be seen that the voltage converges to values close to those of the reference in more than three network cycles. The dc variable cannot completely reach a steady state in the maximum time of the sample. It is also possible to see a small distortion in the load current in transient state, which is when the THDi is at 9.7%.

### 5.2. qZSI Operating under an FCS-MPC Scheme

By forcing the system to operate under the FCS-MPC scheme at all times, it is possible to see that for an equivalent test applied to the converter, capacitor voltage reference step change from 40V to 65V, the response takes a bit longer than a cycle to reach a steady value (see Figure 9).

Taking into consideration the FCS-MPC features, and the fact that the sampling time is only 20kHz, the scheme achieves a satisfactory tracking of the dc voltage reference. However, the system still has a small error in steady state and a THDi of 14.5%, which is greater than the distortion values of the previous controller. This distortion increases if the dc side voltage is increased.

Considering the step change from 65V to 40V under the predictive control scheme, the system reaches steady state in virtually two and a half cycles (Figure 10). A closer analysis reveals a small error in steady state in the dc variable and a greater distortion in the current in relation to the PWM linear scheme.

Looking at Figure 7, Figure 8, Figure 9 and Figure 10, and when considering the THDi values of the current in steady state in all cases, it can be asserted that if an FCS-MPC strategy (with 20 kHz in its sampling frequency) is compared with a PWM linear control using 20 kHz on its carrier frequency, the THDi is lower in the case of PWM linear control. Considering Figure 7, Figure 8, Figure 9 and Figure 10, one can conclude that the most apparent advantage of operating with FCS-MPC is a faster dynamic response against a reference change. For this reason, the ACMA proposal seeks to control the converter using PWM linear control in a stationary condition and an FCS-MPC in a transient state.

### 5.3. ACMA Performance

As previously mentioned, the implementation of a basic criterion such as the one presented in (Equation 26) in the form of an alternating control mode algorithm (ACMA) may have bouncing issues, which can be observed in Figure 11. These bounces appear more noticeably as the ripple in the dc voltage of qZSN capacitors increases. The criterion variable $\eta_{CM}$ oscillates at least six times before reaching a steady state and operating under SCM.

Conversely, Figure 12 implements the proposed ACMA as described in (Equation 27), and with values that are properly selected, it is possible to observe that the scheme does not have bouncing issues or unnecessary changes from one control mode to the other. Once ACMA detects that the system is operating under DCM, it changes to the predictive control scheme, and it returns to SMC when the ACMA indicates it.

Figure 12 shows that the ACMA successfully detects the transient state and activates DCM. Conversely, when the ACMA detects the steady state, it switches to SCM, without showing any bouncing issues. Through this test, which is equivalent to those presented for the schemes operating separately, it is possible to observe that the time the converter takes to move into steady state is a bit longer than a cycle.

On the other hand, Figure 13 shows that for the step test from 65 V to 40 V, with the system operating with ACMA, the transient state lasts fewer than three network cycles. This shows that the implementation of the proposal helps to achieve both the dynamic performance of a predictive control scheme in transient state and those of a PWM linear control scheme in steady state. It may be essential to mention that the ripple percentages in the capacitor $C_{1}$ voltage have been measured at both acquired levels in Figure 13. The ripple percentage in the dc voltage of the capacitor when the average value is 40V is around 17%, while that same measured percentage is 4.76% when the average dc voltage on the capacitor $C_{1}$ is 65V. These measured percentages depend on the load and the controlled level of dc voltage.

Figure 14 shows that during a start-up with null initial conditions, and given the references in the ac and dc sides, the system will start in DCM. In fewer than two cycles, it will reach the desired operating point with 65 V and 1.8 A of amplitude in the current, then the SCM is activated, which allows the converter to work under the PWM linear control scheme. This last test shows the possibly major advantage of the scheme, considering that in a system operating with linear controllers only, other strategies are usually needed to bring the system into a valid operating point.

In summary, for the experimental results, it has been necessary to evaluate the control proposal against a significant load, considering the size of the qZSN filter. Consequently, it has been possible to see that the proposed strategy (in Figure 5) works successfully even in the face of considerable dc ripple oscillations (Figure 12, Figure 13 and Figure 14). On the contrary, if a “basic switching criterion” is implemented (such as the one presented in Equation (Equation 26)), this dc ripple makes the basic strategy act with bounces, similar to what one can see in Figure 11.

### 5.4. Regarding Complexity of the Proposal and Computational Burden

Given that the ACMA algorithm exclusively operates an FCS-MPC or a modulation-based linear controller (never both schemes at the same time), the computational burden is given mainly by the controller that requires the more significant processor requirement (in the case of this work, the predictive algorithm). The proposal’s complexity is (for the person in charge of the implementation) when programming the scheme since it requires a more significant number of lines in the code. However, it is worth noting that, considering today’s digital signal processors (DSP) or system-on-chip (SOC) systems, the proposal does not present any challenges from a hardware point of view.

## 6. Conclusions

PWM linear control and FCS-MPC were compared when applied to an SP-qZSI. Based on experimental results and using the same sampling frequency for both schemes, it is demonstrated that PWM linear control strategy exhibits good performance in steady state, mainly from the point of view of the harmonic distortion in the load current giving a THDi of 5.1%. On the other hand, FCS-MPC performed better during transient and achieved faster dynamic responses than those the linear control, although with evident distortion in the load current, giving a THDi of 14.5%, and a minor tracking error in steady state.

Then, an alternating control strategy was proposed for the SP-qZSI, aiming to combine in one hybrid scheme the best characteristics of each strategy in order to achieve a better performance during transients and in steady state. The proposed strategy uses a PWM linear control strategy to operate as SCM in steady state, and a FCS-MPC to operate as DCM during reference changes. The selection of the control mode was done by an ACMA designed with anti-bouncing capability.

Experimental results demonstrate that the alternating hybrid strategy applied to the SP-qZSI performed better than each strategy operating individually. Results shown that similar distortion to the one obtained with the linear control are achieved but with the dynamic behavior of an FCS-MPC. These experimental results prove the feasibility and better performance of the proposed control scheme.

## Figures and Tables

**Figure 2 sensors-22-02458-f002:**
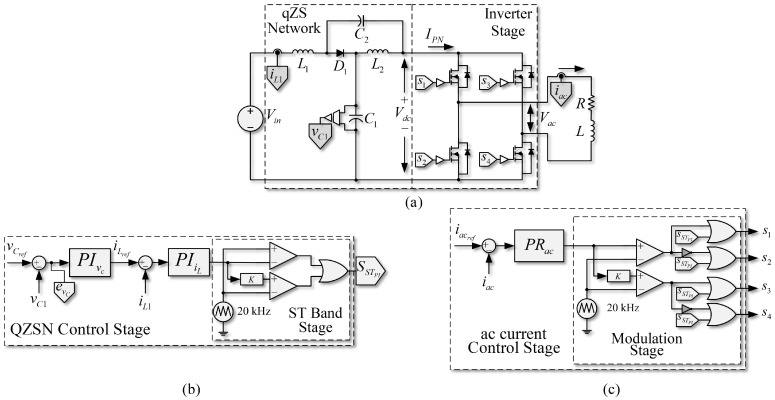
PWM linear control scheme, based on PI controllers for dc-link and PR controller for the ac load. (**a**) Quasi-Z-source inverter with RL load; (**b**) PWM control strategy on the dc side; (**c**) PWM control strategy on the ac side of the inverter.

**Figure 3 sensors-22-02458-f003:**
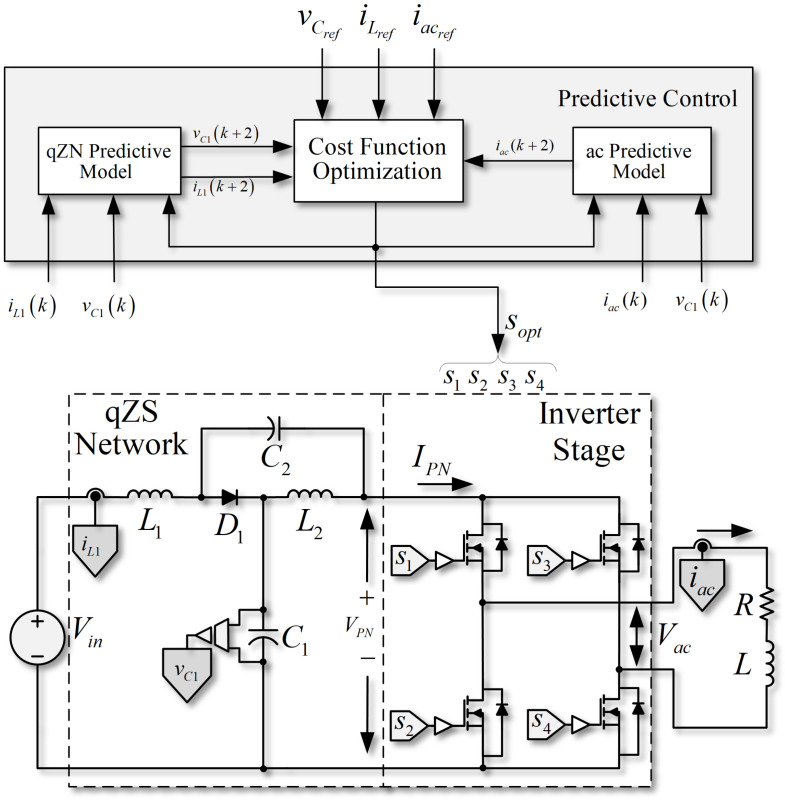
Finite control set model predictive control scheme in an SP-qZSI.

**Figure 4 sensors-22-02458-f004:**
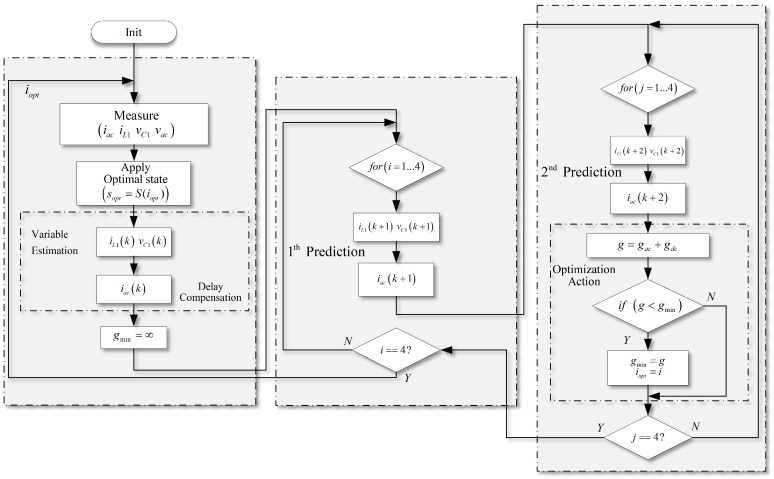
Flowchart of proposed model predictive control for SP-qZSI.

**Figure 5 sensors-22-02458-f005:**
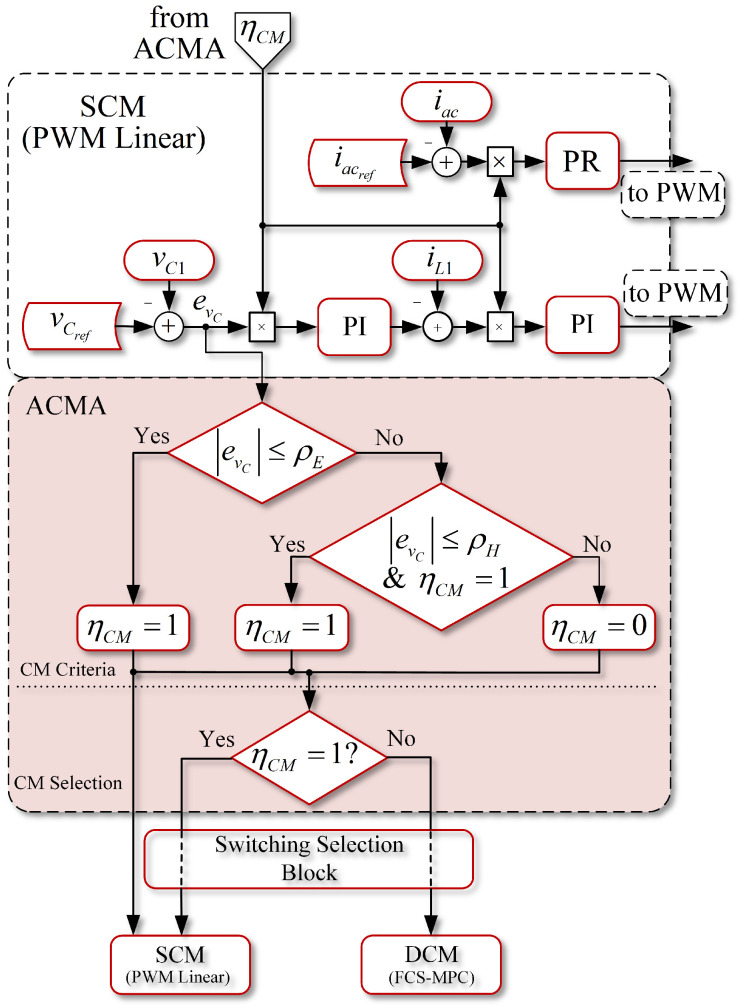
Alternating control mode algorithm flowchart and interaction with SCM.

**Figure 6 sensors-22-02458-f006:**
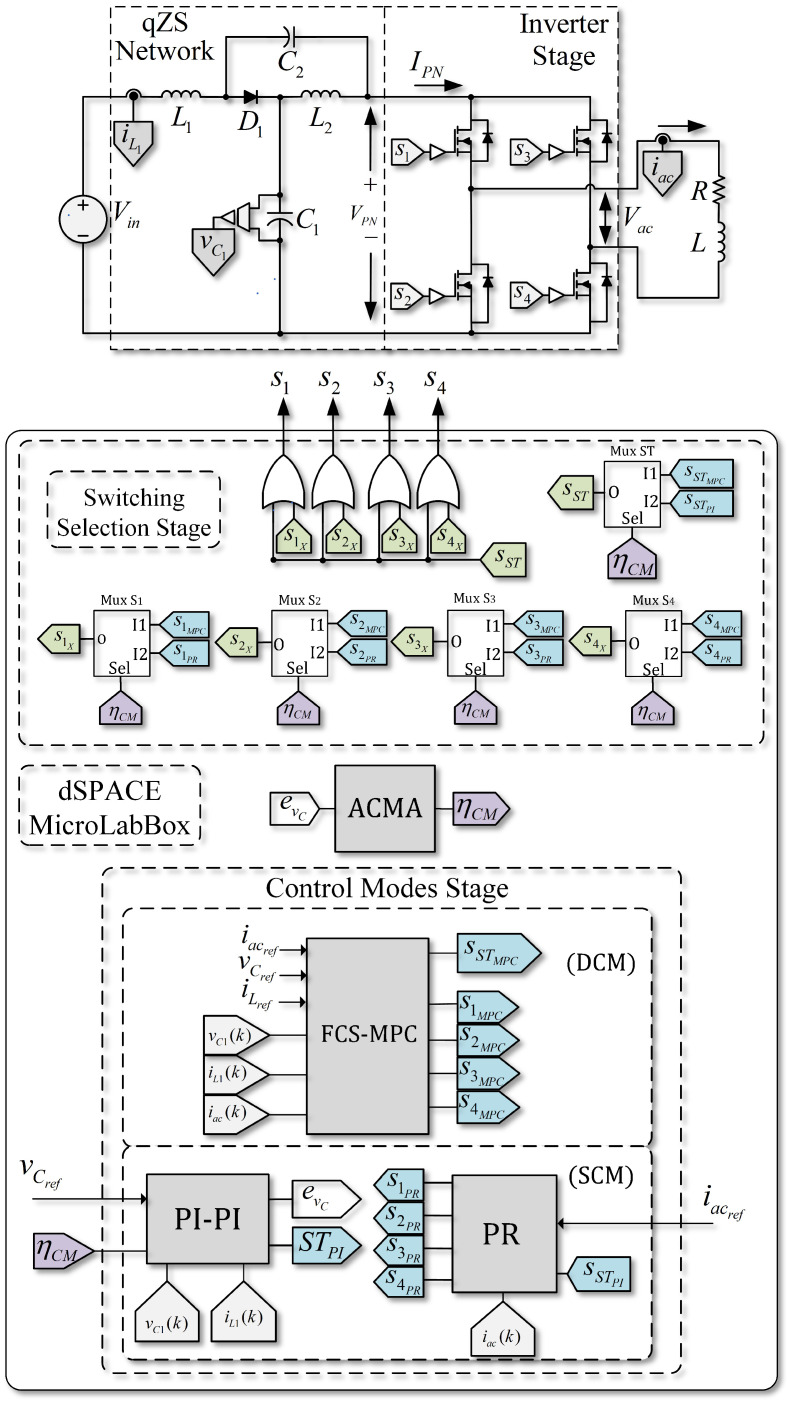
Alternating control scheme proposed, based on linear and FCS-MPC strategies.

**Figure 7 sensors-22-02458-f007:**
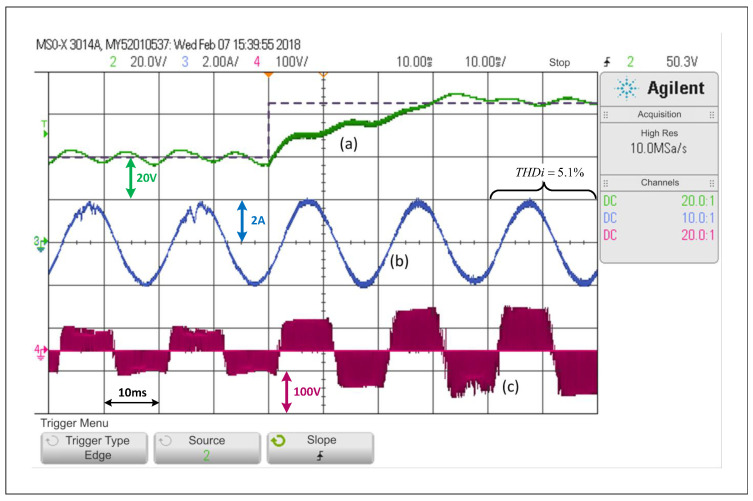
Response of the PWM linear control scheme in the SP-qZSI in the face of a step reference from 40V to 65V, (**a**) dc voltage variable response, (**b**) ac load current response, (**c**) inverter output voltage.

**Figure 8 sensors-22-02458-f008:**
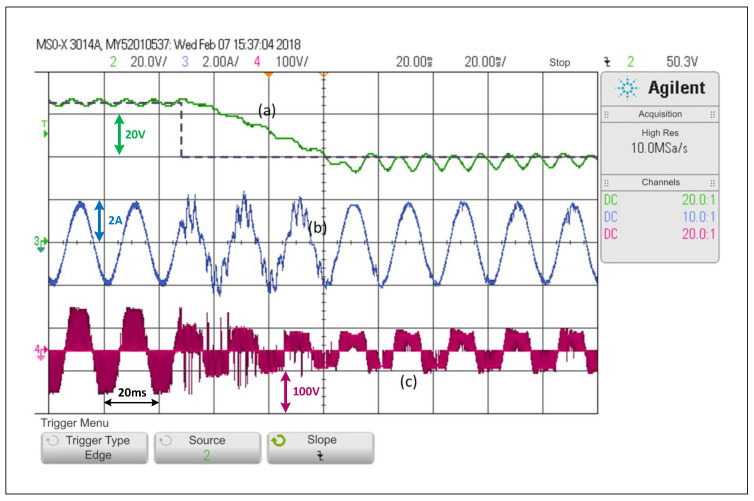
Response of the PWM linear control scheme in the SP-qZSI in the face of a step reference from 65V to 40V for the dc voltage, (**a**) dc voltage variable response, (**b**) ac load current response, (**c**) inverter output voltage.

**Figure 9 sensors-22-02458-f009:**
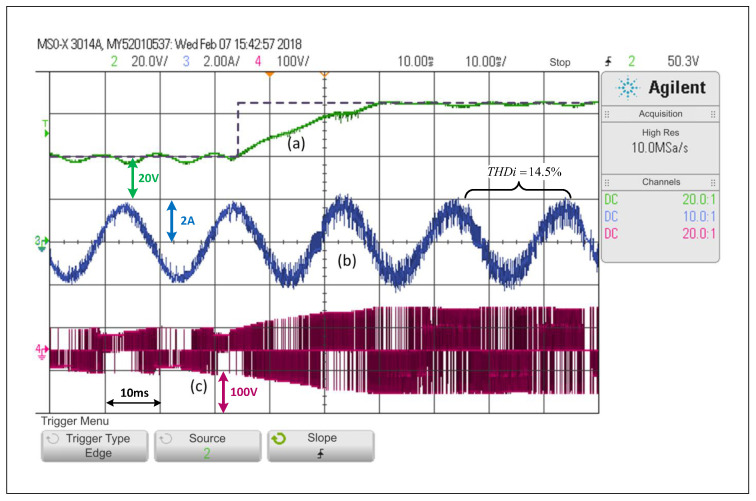
Response of the FCS-MPC scheme in the SP-qZSI in the face of a step reference from 40V to 65V, (**a**) dc voltage variable response, (**b**) ac load current response (**c**) inverter output voltage.

**Figure 10 sensors-22-02458-f010:**
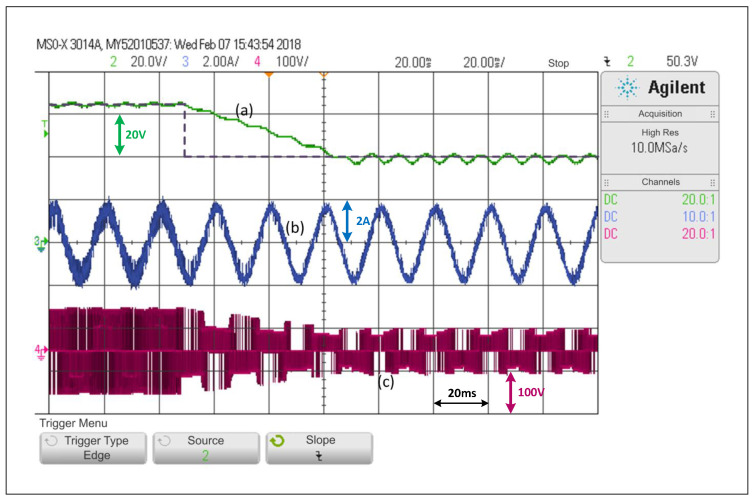
Response of the FCS-MPC scheme in the SP-qZSI in the face of a step reference from 65V to 40V: (**a**) dc voltage variable response, (**b**) ac load current response, (**c**) inverter output voltage.

**Figure 11 sensors-22-02458-f011:**
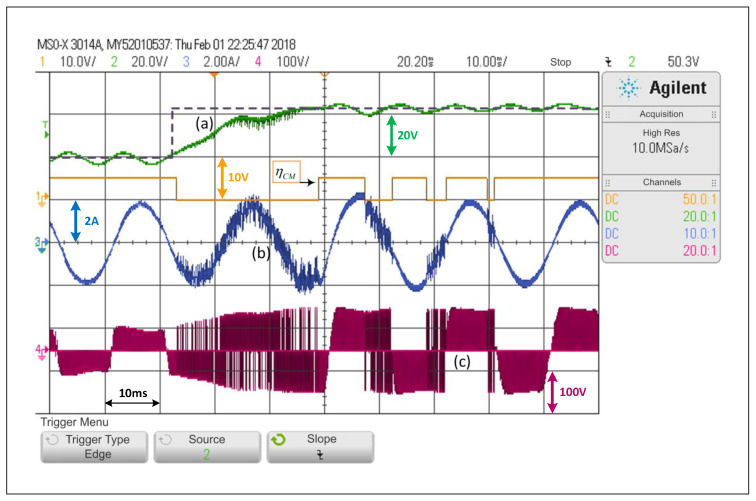
Response of the alternating control scheme with basic criterion in the SP-qZSI with a step reference from 40V to 65V: (**a**) dc voltage variable response, (**b**) ac load current response, (**c**) inverter output voltage.

**Figure 12 sensors-22-02458-f012:**
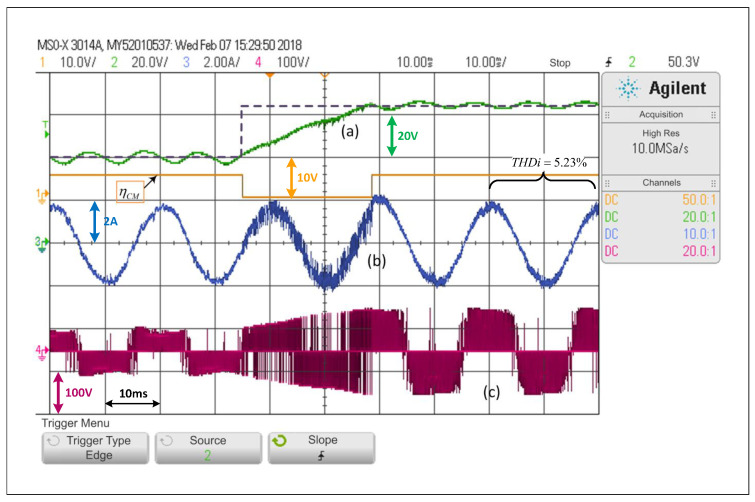
Response of the alternating control scheme with ACMA proposed in the SP-qZSI with a step reference from 40V to 65V: (**a**) dc voltage variable response, (**b**) ac load current response, (**c**) inverter output voltage.

**Figure 13 sensors-22-02458-f013:**
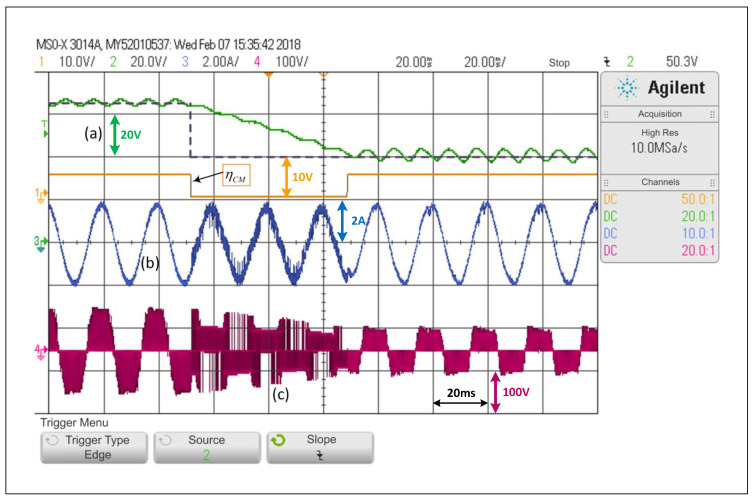
Response of the alternating control scheme with ACMA proposed in the SP-qZSI with a step reference from 65V to 40V: (**a**) dc voltage variable response, (**b**) ac load current response, (**c**) inverter output voltage.

**Figure 14 sensors-22-02458-f014:**
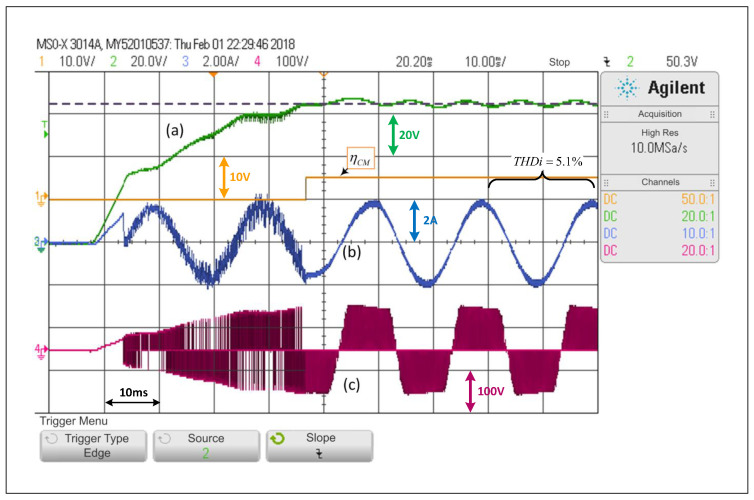
Response of the alternating control scheme with ACMA proposed for SP-qZSI in a start-up, 0 to 65V in capacitances and 0 to 1.8 A in the load: (**a**) dc voltage variable response, (**b**) ac load current response, (**c**) inverter output voltage.

**Table 1 sensors-22-02458-t001:** qZSI valid states.

Index	$s_{1}$	$s_{2}$	$s_{3}$	$s_{4}$	$S_{f}$	$S_{\mathrm{ST}}$	State	$V_{\mathrm{PN}}$	$V_{\mathrm{ac}}$
1	1	0	0	1	1	0	nSTS	$v_{C1}+v_{C2}$	$v_{C1}+v_{C2}$
2	0	1	1	0	−1	0	nSTS	$v_{C1}+v_{C2}$	$-(v_{C1}+v_{C2})$
3	1	0	1	0	0	0	nSTS	$v_{C1}+v_{C2}$	0
	0	1	0	1	0	0	nSTS	$v_{C1}+v_{C2}$	0
4	1	1	1	1	0	1	STS	0	0

**Table 2 sensors-22-02458-t002:** Setup parameters.

Variables	Description	Values
$V_{in}$	Source voltage	30V
$L_{1}$ & $L_{2}$	qZSN inductors	1.5mH
$C_{1}$ & $C_{2}$	qZSN capacitors	470μF
$R_{L}$	Resistance load	17Ω
$L_{L}$	Inductor load	25mH
$f_{0}$	Output frequency	50Hz
$v_{ref1}$	Voltage reference 1st step	40 V
$v_{ref2}$	Voltage reference 2nd step	65 V
$i_{acref}$	Current ac side reference	1.8 A

**Table 3 sensors-22-02458-t003:** Control scheme parameters.

Variables	Description	Values
$f_{s}$	Sampling frequency for FCS-MPC and PI-PI-PR	20kHz
$f_{PI+PR}$	PWM carrier frequency for PI-PI-PR	20kHz
$\lambda_{i_{L}}$	Weighting factor qZSN. Inductor current prediction	1
$\lambda_{v_{c}}$	Weighting factor qZSN. Capacitor voltage prediction	1.2
$\lambda_{i_{ac}}$	Weighting factor ac load current prediction	0.45
$K_{Pv}$	Proportional gain capacitor voltage loop	0.9
$K_{Iv}$	Integral time capacitor voltage loop	0.02 s
$K_{Pi}$	Proportional gain inductor current loop	−1.3
$K_{Ii}$	Integral time inductor current Loop	0.0005 s
$K_{Pr}$	Proportional gain PR loop	100
$K_{R}$	Resonance gain PR loop	800
$\omega_{0}$	Resonance frequency	2π50rad/s

**Table 4 sensors-22-02458-t004:** ACMA parameters.

Variables	Description	Values
$\rho_{E}$	Error reference band	3V
$\rho_{H}$	Error reference hysteresis band	6V

## Data Availability

Not applicable.

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
