# Peer review of "Application of a Control Scheme Based on Predictive and Linear Strategy for Improved Transient State and Steady-State Performance in a Single-Phase Quasi-Z-Source Inverter"

_sensors, 2022, doi:10.3390/s22072458_

Round 1

Reviewer 1 Report

  1. Recent literature is not studied properly
  2. What is original contribution? Highlight in introduction.
  3. Analysis is not strong. Need to be evaluated how switching optimization can be done. How to ensure reduced/constant switching frequncy?
  4. Please derive the achievable gain with this converter and control scheme?
  5. Why choosing predictive control over conventional PI controller? How much improvement has been reported with the proposed scheme need to be highlighted.
  6. Why DC-link voltage is having significant variations irrespective of the input filter?
  7. Experimental results are not satisfactory. Seems proper switching is not happening with the proposed controller.
  8. Computational complexities need to be evaluated with appropriate comparative study.

Author Response

Dear Reviewer

I hope you are well.

I am submitting a new version of the paper. In addition to the paper's changes (according to your requirements), I am attaching a response letter for you in a pdf file.

Best regards

Reviewer 2 Report

The static power converters have a wide range of industrial applications, which is why the topic addressed by the authors of the paper is very important. Their efficient control also offers the possibility of achieving high energy efficiency. I believe that by publishing the paper, important scientific results are made available to researchers in this field.

My comments on this paper are:

  1. In Figures 7,…, 14 it is necessary to translate the information into English, respectively it is necessary that the colors in the image presented to the right of the figure are also found in the colors of the sizes represented in these figures.
  2. In Figures 7,…, 14 it is necessary to specify which size represents the ordinate and which are the representation scales.
  3. In figure 1a are the coils having the inductances L1, respectively L2 have iron core or not? If they have an iron core, then it must be taken into account that active power losses occur in the magnetic core, which causes the electrical resistance of the coils to obtain high values, which is why neglecting these resistances produces significant errors, which can no be neglected.
  4. Lines 66-67 after: the sign must be entered; nu and.
  5. In Figure 1 are capacitors C1 and C2 electrolytic? If they are electrolytic, the polarity of the voltage (+) cannot be changed to (-).
  6. If the inductors L1 and L2 respectively are not considered ideal, the equations 1,…, 11 change, and as a result the relations 12, 13, 14 change. Can you appreciate the value of the errors introduced by neglecting the electrical resistances of the two coils?
  7. The consumer current in Figure 9 is more distorted than in Figure 7, shouldn't it be the other way around? In figure 8 and figure 10 it is the other way around, why?
  8. The paper presents the total distortion coefficients of the load current for the existing situation, but does not present the values of these coefficients in the scheme proposed by the authors. In order to justify that the proposed scheme is more efficient, it is necessary to compare the distortion coefficients obtained with the proposed scheme with those obtained in the existing schemes.

Author Response

Dear Reviewer

I hope you are well.

I am submitting a new version of the paper. In addition to the paper's changes (according to your requirements), I am attaching a response letter for you in a pdf file.

Best regards.

Reviewer 3 Report

Authors should make the following corrections

Figures 2, 5 and 6 must be placed after being described

Tables 1, 2 and 3 must be placed after being used

The conclusions must be validated with data obtained in the results. In addition, it should be clarified that in the results they indicate that THDI is not adequate and in the conclusions they say that the THDI obtained is good, which is contradicted.

Author Response

(The authors gave the same response as above.)

Round 2

Reviewer 1 Report

  1. In line with previous comment 4: Additional analysis is essential for the qZS gain vs. duty ratio. Is there any condition for the maximum gain of the overall converter?
  2. The response to previous comment 6 is not satisfactory. In all experimental results, why is DC-link voltage having significant ripple? Please provide an appropriate explanation to this. Also, improve all the experimental results with minimal ripple in DC-link voltage.
  3. In line with previous comment 7, please provide the experimental results of average switching frequency for inverter as well as qZS network considering the different load capability.
  4. Still, much language improvements are necessary. The paper flow is not proper for this article, which needs an appropriate proofread before resubmission.

Author Response

Dear Reviewer

I enclose a response letter.

Your review is appreciated.

Best Regards.

Reviewer 3 Report

The authors must place the figures after being mentioned in the text to understand the explanation given for each of them.

Author Response

Dear Reviewer

We have made the adjustments you requested.

Other aspects of the job have also been improved in the new version of the paper.

Your review is appreciated.

Best regards